# Application of Acellular Dermal Matrix in Gynaecology—A Current Review

**DOI:** 10.3390/jcm11144030

**Published:** 2022-07-12

**Authors:** Kaja Skowronek, Wojciech Łabuś, Rafał Stojko, Diana Kitala, Marcin Sadłocha, Agnieszka Drosdzol-Cop

**Affiliations:** 1Chair and Department of Gynecology, Obstetrics and Gynecological Oncology, Medical University of Silesia in Katowice, Markiefki 87, 40-211 Katowice, Poland; rafalstojko@gmail.com (R.S.); marcin-sadlocha@wp.pl (M.S.); cor111@poczta.onet.pl (A.D.-C.); 2Dr. Stanislaw Sakiel Centre for Burn Treatment in Siemianowice Slaskie, Jana Pawla II 2, 41-100 Siemianowice Slaskie, Poland; wojciech.labus@gmail.com (W.Ł.); diana.kitala@clo.com.pl (D.K.)

**Keywords:** acellular dermal matrix, pelvic organ prolapse, reconstructive gynecology, vaginal reconstruction, sacrocolpopexy, AlloDerm transplant, vaginal mesh

## Abstract

The aim of our study is to draw attention to the multitude of applications of acellular dermal matrix (ADM) in the surgical treatment of urogynaecological disorders, such as reduction in the reproductive organs, and in reconstructive gynaecology. Despite the existence of numerous operational methods and materials, the effectiveness of transvaginal operation is still insufficient. Native tissue operations are often not durable enough, while operations with synthetic materials have numerous side effects, such as infections, hematomas, vaginal erosion, or dyspareunia. Hence, the search continues for a different material with a better efficacy and safety profile than those previously mentioned. It seems that ADM can meet these requirements and be a useful material for urogynaecological surgery. Key words related to the usage of ADM in gynaecological reconstructive surgery were used to search relevant databases (NCBI MedLine, Clinical Key, Clinicaltrials.gov). This manuscript is based on 43 literature sources, 28 (65.11%) of which were released after 2016. Older sources are cited for the purpose of presenting basic science, or other important issues related to the manuscript. ADM seems to be an ideal material for urogynaecological and reconstructive surgery. It has high durability, and thus high effectiveness. Moreover, it does not have the side effects typical for synthetic materials. There are no reports of material rejection, erosion or dyspareunia directly related to the presence of the mesh. Due to the difficulties in obtaining ADM and the need to perform additional tests, this material is not common in routine clinical practice. Therefore, the number of cases and the size of the research groups are insufficient to clearly define the potential of mesh from biological tissue. However, the results are so promising that it is worth considering a wider introduction to the use of this material. Our hope is that increasing clinicians’ awareness of this topic will lead to more studies comparing methods using native tissues or synthetic materials and those using ADM.

## 1. Introduction

The aim of our study is to draw attention to the multitude of applications of acellular dermal matrix (ADM) in the surgical treatment of urogynaecological disorders, such as reduction in the reproductive organs, and in reconstructive gynaecology. Urogynaecological disorders have both medical and social consequences. Patients complain about physical and mental ailments, limited sexual function, and the consequent reduction in quality of life [1].

This problem is also significant for the healthcare system. According to sources, in developed countries, operations for static disorders in the reproductive organs account for 20% of all gynaecological operations. These ailments may affect 50% of women who give birth. Eleven percent of women under 80 years of age will undergo surgery for varying degrees of pelvic organ prolapse (POP) or urinary incontinence. Every third patient will require more than one operation [1]. Surgical treatment may involve the use of the patient’s autologous native tissue, the implantation of synthetic prostheses, or most often polypropylene or biological materials [1,2]. The next group are cancer patients who, as a result of aggressive treatment for the underlying disease, gain an opportunity for a longer life, but not without the complications of this treatment.

Recently, synthetic materials have not been used very often in urogynaecology due to warnings from the Food and Drug Administration (FDA). Although many sources emphasize that a significant part of the complications experienced result from wrong qualification or the wrong selection of an implant for a static defect, the interest in this method has weakened. The most common complications after this type of surgery include the erosion of the mesh into the vagina, bladder or rectum, the recurrence of prolapse, pelvic pain syndrome, dyspareunia, vaginal stenosis and/or shortening, and postoperative fistulas [1,2,3].

In order to exclude unfavourable phenomena related to the use of synthetic materials, the use of biological materials is often proposed. In this perspective, cell-free tissue transplants are of great interest. One is the acellular dermal matrix (ADM). This is a promising material that has a chance to replace previously used synthetic materials. Due to its biocompatibility, we can expect significantly fewer side effects [4,5,6,7,8].

The essence of the method is the use of the extracellular matrix of the allogeneic dermis as a scaffold for mechanical protection against in vivo forces until the transplanted ADM becomes an integral part of the body. This can be considered the main advantage of this technique [8,9,10,11]. The skin is a collagen-rich tissue and is a rich source for biomaterials used in tissue engineering [11]. By removing cells from the allogeneic human dermis, a cell-free, collagen-free, non-immunogenic mesh is obtained that can be de novo revitalized by autologous cells. Such an acellular skin matrix (ADM) or acellular skin graft (ADG) can be a stimulus for natural mechanisms of regeneration and reconstruction [4,5,6,7,8,11]. Such ADMs are created mainly in a multi-stage application process, e.g., with proteolytic enzymes on allogeneic human skin that have been retrieved from a deceased donor [4,7,11] (Figure 1).

## 2. Materials and Methods

Key words related to the usage of ADM in gynaecological reconstructive surgery (such as acellular dermal matrix, reconstructive gynaecology, AlloDerm, and vaginal mesh) were used to search relevant databases (NCBI PubMed, Clinicaltrials.gov, accessed on 21 April 2022).

The inclusion criteria were papers in Polish and English, the results of the research and clinical trials, and case reports describing the use of ADM in urogynaecological surgeries and reconstructive gynaecology. We rejected papers that were based on other materials that were not compared to ADM, or were beyond our chosen medical field, such as ophthalmology, wound care, and breast diseases.

This manuscript is based on 43 literature sources, 28 (65.11%) of which were released after 2016. Older sources are cited for the purpose of presenting basic science or other important issues relevant to the manuscript.

## 3. Results

The immune response is mainly directed against the proteins and lipids of cell membranes; thus, cell extraction from tissue is a promising method to avoid the creation of a post-transplant immune response in the body of the recipient [4,12,13,14,15,16]. The removal of cellular components should minimize the immunologically induced inflammatory process, which may weaken the biodegradation process of the transplanted bioprosthesis [15]. The procedure of removing cells from allogeneic human dermis results in a cell-free, non-immunogenic matrix composed of extracellular matrix (ECM) elements that can be re-inhabited by autologous cells. Such cell-free dermal matrices (ADMs) can also stimulate natural regenerative and reconstructive mechanisms [12,13,14,15,16,17,18,19].

Based on the results of numerous studies, it has also been shown that cell-free ECM scaffolds are able to propagate and support the growth and differentiation of many types of cells in vitro and induce constructive tissue remodelling processes after in vivo transplantation [7,8,9,10,11]. Thus, the use of three-dimensional ECM scaffolds obtained from tissues or whole organs that have undergone cell removal procedures is an increasingly used strategy of regenerative medicine and tissue engineering [4,5,11,12,13,14,15,16,17,18].

### The Use of ADMs in Gynaecological Reconstructive Surgery

The safety and efficacy of this biological material have been confirmed in animal models. In 2009, Zhou et al. conducted a study on 12 guinea pigs in which animals underwent resection surgery followed by vaginal reconstruction using ADM. At a later stage, the animals were sacrificed at various intervals from the procedure, and then the tissues were subjected to immunohistochemical staining and Van Gieson staining. This made it possible to study the growth of individual layers of the vagina. Scientists checked the presence of epithelial and smooth muscle tissue and the contractile activity of the isolated vaginal smooth muscle. Based on these studies, the epithelization of two-thirds of the new vaginal mucosa was observed after just one week. One to two layers of epithelium were visible at this stage. Within 4–6 weeks, the epithelization of the saliva increased to 4–5 layers and single smooth muscle cells appeared. After 12 weeks, the produced vagina had a normal structure, indistinguishable from the native tissue [20].

Another study that has confirmed the effectiveness of ADMs in gynaecological reconstructive surgery was performed by Peró et al. [20,21]. The main aim of Peró et al.’s study was to assess the usefulness of New Zealand white rabbits (NZW) as an animal model for research on the use of biomaterials in pelvic reconstructive surgery. The second aim was to compare the complications of using a standard polypropylene mesh and human acellular human matrix (hADM). The materials were implanted in the animals subcutaneously, into the abdominal wall and into the vaginal submucosa. After 180 days, the grafts were removed. Experience has shown that hADM is associated with a lower frequency of clinical complications and better macroscopic tissue integration compared to synthetic mesh. The abnormalities included vaginal mesh extrusion of 33% for PP vs. O% for hADM (*p* = 0.015) [2].

Additionally, previous studies suggest the advantage of biological materials over synthetic ones in terms of their impact on the tissues of patients. Marc Gualtieri et al. published a study comparing the effect of propylene mesh and porcine dermal acellular collagen matrix mesh with and without estradiol supplementation on vaginal smooth muscle cells (VaSMC). The proliferation of VaSMC in the porcine dermal acellular collagen matrix was higher than in the polypropylene mesh-exposed cells. Relative cell numbers in first group were 0.27 ± 0.03 vs. 0.21 ± 0.01, *p* = 0.03, in the second group. The results confirmed this advantage. Estradiol supplementation increased this effect. This may explain the reduced number of complications for mesh erosion with biological meshes compared to propylene meshes [21].

Said et al. described, in 2007, a case of the successful reconstruction of the pelvic floor and perineum using HADM and femoral lobes after pelvic exenteration and radical vulvectomy in a 75-year-old woman with recurrent squamous cell carcinoma of the vulva and osteoradionecrosis. Despite the large area of the defect, radiotherapy, and bacterial infection, the operated site healed properly. The choice of biological material was made due to the lower percentage of complications such as adhesions, infections and erosions [22].

Almost 10 years later, Bhavsar et al. described the case of a 51-year-old female patient with diagnosed rectal adenocarcinoma and performed an abdominal–perineal resection (APR) and end colostomy, in which the posterior vaginal wall was excised and a large yawning defect was formed. The patient had been treated in the past with radiotherapy for cervical cancer. As a result, she developed a slight narrowing of the vagina, which made it impossible to recreate the continuity of the vagina with the help of side fragments. It was decided to create a vagina with Alloderm^®^. An acellular collagen matrix was customized into a sheet 8 cm in length and 4 cm in width and stabilized with 3-0 Vicryl sutures. A Foley catheter was inserted into the uterine cavity to maintain its patency. The postoperative course of the patient was normal. Ten months after the surgery, vaginoscopy was performed, showing 7 cm of vaginal vaults without strictures and cavities. The posterior wall was properly healed with no morphological differences from the native tissue. Three years after the operation, the patient reported vaginal scar pain. The examination showed partial stenosis of the vaginal vault and atrophic mucosa. This condition could have been influenced by both the use of a specific material and the individual properties of the patient’s organism. Although the authors emphasize the need for further research, they suggest that allogeneic transplantation enables vaginal reconstruction after oncological operations in the anorectal area while maintaining anatomy and good functional results. This method also shortens the time of surgery and perioperative morbidity, allowing the patient to avoid additional procedures such as autologous transplants or the reconstruction of the intestine [23].

Similar conclusions were reached in 2007 by researchers using HADM to repair vesico-vaginal fistulas and reconstruct the urethra in patients with urethral stricture. The operator used retroperitoneal access through the bladder in these surgeries, and then used HADM as a tissue patch to repair four complicated cases of fistulas. Very good postoperative results were obtained. No postoperative complications were reported. Urine leakage was stopped immediately after surgery. No patient reported a recurrent leakage of urine during the 4–12-month follow-up. Urethrography revealed the excellent calibre of the operated urethra [24].

Another report on ADM in reconstructive gynaecology by You et al. compared the effects of standard vaginal-rectal fistula treatment and the treatment of this disease using a tissue patch. The study included a small study group (12 people) and a control group of 10 people. In 11 out of 12 people, the first operation brought full therapeutic success, and in 1 of them reoperation was necessary. Based on the results, the authors of the study concluded that the use of ADM is associated with greater effectiveness and lesser traumatization of the patient [25].

In 2019, Wang et al. published a study on the effectiveness of vaginoplasty using ADM. The study group included 16 patients with early-stage cervical cancer who were treated with surgery and radiotherapy. They underwent vaginoplasty using ADM and were then advised to use vaginal dilators for six months. The effectiveness of the treatment was assessed after 12 months. The surgical method turned out to be safe, and no intraoperative complications were reported. The vaginal width increased significantly from 1.31 ± 0.4 cm before the procedure to 4.13 ± 0.43 cm after the procedure (*p* = 0.034). The vaginal length also increased from 5.97 ± 0.59 cm to 9.25 ± 0.66 cm (*p* < 0.001). Most of the patients (75%) reported a satisfactory sex life [26].

Five years earlier, Zhu et al. applied ADM to vaginoplasty in 53 patients with Mayer–Rokitansky–Küstner–Hauser syndrome. As in the above-mentioned study, patients were instructed to use a vaginal dilator postoperatively. The control group consisted of healthy women of the same age as the study group. The length of the vagina ≥8 cm and the width of at least two fingers was determined as an anatomical success. Sexological results were assessed using the following standardized questionnaires: the Female Sexual Function Index (FSFI) and the Body Image Perception Questionnaire. The examination did not reveal any significant intraoperative complications, infection, rejection or detachment associated with transplantation. Postoperative vaginal granulomatous polyps requiring outpatient removal developed in 11.3% of the patients. During follow-up (mean time 21.1 months), the anatomical success was 100%. Among the sexually active group of 32 patients, 75% achieved a result in the FSFI questionnaire similar to the control group (26.7 ± 3.5 vs. 25.6 ± 7.4, *p* = 0.46) [27].

In the literature, we also found a case of neovagina formation in a 33-year-old patient with MRHK syndrome, in whom the classic method could not be used due to severe pain and lichen sclerosus [28].

The use of ADM in gynaecological surgery is not limited to reconstructive surgery only. Matrices are also used in the correction of defects in the anatomy of the reproductive organ. At Peking University People’s Hospital, 20 patients with the problem of the prolapse of the reproductive organ underwent surgery using acellular dermal mesh. All patients had an anterior vaginal descent and 17 of them also had a posterior vaginal wall. Fifteen patients were implanted with ADM on the anterior wall, two on the posterior wall, and three patients on both walls. During an average observation of 9.3 months (6 to 12 months), no erosion or infection was found. There were four cases of recurrence (20%) approximately six months after the surgery. Three were related to the reduction of the first degree, and one of the second degree. The patients did not report any other complaints. The patients’ tolerance was described as good, and the use during surgical treatment was easy and simple [29].

The research team of Ward et al. performed a retrospective analysis on thirty-three women suffering from recurrent stage II or primary or recurrent stage III-IV anterior vaginal dislocations who had undergone paravaginal vaginal reconstruction using the commercially available human cell-free skin matrix AlloDerm in the period from November 1998 to April 2002. The surgical technique consisted of opening the anterior vaginal wall in the midline from the apex of the vagina to 2 cm near the external opening of the urethra. The cleavage was paravaginal along the levator ani muscles. Local defects in the cup-ion fascia were repaired with interrupted 2-0 polyglycolic acid sutures. The 3 × 7 cm AlloDerm was cut into a trapezoidal shape and specially tailored to each patient individually so that the proximal graft edge was 6–7 cm long, the distal edge was approximately 4 cm, and the height was 3 cm. The grafts were positioned above the base of the bladder and sewn to the fascia of the arcus tendineus of the pelvis with four interrupted 2-0 sutures (braided durable polyester stitching on both sides) [16].

Out of 33 women who underwent the described reconstructive procedure, 20 were able to undergo regular annual clinical follow-up. After trying to contact all women, long-term data were obtained for 24 out of 33 patients (72.7%). Of the remaining nine women, three were contacted by phone, but were unable to attend, one died, and five were not contacted. All three women contacted by phone denied symptomatic prolapse and had no other urogynological complaints. The women did not seek urogynological care from other doctors. To summarize, paravaginal reconstructions using the AlloDerm matrix turned out to be safe and well tolerated by the recipients [19].

Earlier observations of this group were made by Clemons et al. Thirty-three women who underwent paravaginal vaginal reconstruction using the AlloDerm transplant due to anterior vaginal wall prolapse were analysed. Preoperatively, 6 women had recurrent stage II, 24 women had stage III, and 3 had stage IV. After surgery, 12 patients had asymptomatic stage II, and 1 was symptomatic. Complaints about problems with urination were resolved in 11 of 14 women (79%, *p* = 0.004), and urinary incontinence in 20 of 23 (87%, *p* < 0.001). Twenty-one women from the study group were sexually active and none of them complained of dyspareunia. Complications included one febrile infection, one cystotomy, and one anterior vaginal hematoma due to heparin therapy. No cases of material erosion or ejection were observed [30].

hADM has also found application in laparoscopic surgery. In 2019, Karon and Chatterjee conducted a study on the effectiveness and satisfaction of patients after laparoscopic sacrocolpopexy using a non-crosslinked acellular dermal matrix. Sacrocolpopexy is the method of choice for the surgical treatment of the recurrent prolapse of the reproductive organ [31,32]. The records of 211 patients who underwent surgery in the period between 2012 and 2017 were analysed retrospectively. One hundred and five responded to the following surveys: the Pelvic Floor Distress Inventory (PFDI-20) and the Pelvic Floor Impact Questionnaire (PFIQ-7). Most of the studied patients reported an improvement in functioning, assessing it as “a little better” or “much better” (87.5%). Five patients required reoperation (4.76%). The complaints reported after surgery were mainly symptoms of overactive bladders and vaginal discharge, i.e., complications typical of this surgical method. The authors assessed ADM as a good alternative to synthetic materials in sacrocolpopexy surgery [32].

Recent reports on the use of this material include the treatment of vaginal laxity in combination with the use of enriched platelet therapy (EPT). In 52 patients, a U-shaped hADM band was placed by submucosal puncture and EPT was injected three times. The first time was during the procedure, and then the following doses were every month. The effects were assessed six months later using the Female Sexual Function Index and Vaginal Health Index (VHI) scores. Patient satisfaction was measured using the Visual Analogue Score (VAS). Therapeutic success was observed in the form of an increase in the perineal height from an average of 1.5 to 2.2 cm and the visual closure of the vaginal vestibule through the labia minora. Sexual function (7.95 vs. 30.09, *p* < 0.001) and patient satisfaction (11.2 ± 3.3 vs. 19.6 ± 4.1, *p* < 0.001) also increased when assessed. There was an improvement in vaginal flexibility, contractility and hydration. No serious side effects were reported [33].

## 4. Discussion

The use of natural biomaterials is a rapidly developing area of regenerative medicine. Moreover, thanks to the implementation of procedures related to the decellularization of natural tissues and organs, significant progress has been achieved in the therapeutic processes of numerous treatments [4]. From this perspective, acellular dermal matrices (ADMs) deserve special attention. Preparations of this type are used in numerous fields of regenerative medicine. These treatments include, among others, burn treatments, hernia repair, breast reconstruction, etc. [34,35,36,37,38,39,40,41,42,43]. Moreover, there are several reports on the use of ADMs in gynaecological reconstructive surgery [20,21,22,23,24,25,26,27,28,29,30,31,32,33].

The literature consistently reports on the effectiveness of using ADMs. They are very well tolerated by patients and have fewer side effects. In addition, they are characterized by very good efficiency and biocompatibility, which makes them an ideal material for gynaecological surgery. In the case of reconstructive surgery, an additional advantage of allogeneic ADM is the reduction in time and the reduction in the number of procedures, such as autologous transplant collection [23]. Although more than twenty years have passed since the first attempts to use biological matrices in gynaecological surgeries, there are no reports of long-term complications occurring in patients who have undergone such procedures.

Attention is drawn to the fact that there is a shortage of recent literature reports on the use of ADM in gynaecology. A large part of the analysed literature was published before 2015 [19,20,21,22,24,25,26,27,28,29]. Due to the proven effectiveness of the use of ADM in gynaecology, we encourage further research in this area.

The use of synthetic meshes in gynaecological operations is associated with numerous controversies. It is known that the FDA has issued a warning on these products, but standard operations using native tissues tend to be insufficiently effective [44]. The most common complications observed after the use of synthetic materials in the vagina are infections, hematomas, vaginal erosion, and dyspareunia [44,45,46,47,48]. Due to the colonization of the vagina by bacteria, it is not possible to insert the material in a sterile manner. The structure of the mesh may favour colonization by bacteria. This increases the inflammatory response and may accelerate erosion [45,47,48]. Under the influence of the scarring process, the mesh shrinks, which often causes dyspareunia. The material from which the nets are made is inert to the body, and can potentially be broken down into toxic compounds that potentiate inflammatory reactions [45].

However, in most studies, there is a limitation in the small size of the studied groups. Single case reports constitute a large part of the literature. The studied groups are also often very heterogeneous. A limitation that causes the lack of popularity of ADM in routine clinical practice is probably also the acquisition of tissues to produce a matrix. Tissues require special processing and then many additional tests, such as bacteriological and virological tests. The age and condition of the donor may also potentially affect the properties of the resulting matrix [1]. Moreover, there is no doubt that a very important aspect of using ADMs is their safety, which results from an appropriate quality assurance system during production. There are proposals for acceptable standards, e.g., for the content of cell membrane residues, genetic material, etc., remaining after the decellularization process to be defined. With these standards, produced ADMs will be safe for the recipient and fulfil their expected functions [4].

There are no studies that clearly compare the effectiveness and complications related to the available surgical methods in urogynecology. Maher et al., in their analysis of data from the Cochrane Incontinence Group, compared the results of randomized controlled trials of different types of vaginal repair (mesh, biological graft, and native tissue). The results confirmed the greater effectiveness of synthetic meshes in relation to native tissues. The awareness of recurrent genital prolapse in women after mesh application was 10 to 15%, while in patients after surgery on native tissues the percentage was 19%. However, 8% of patients required a second treatment to expose the mesh. Stress urinary incontinence and bladder damage during surgery were more common in this group. There were no differences between the groups in terms of the incidence of dyspareunia de novo. There was no evidence of differences between biological meshes and surgeries on native tissues in terms of reoperation and prolapse awareness. These results should be approached with caution as the results of these studies were defined as low-quality evidence [44]. Studies comparing biological and synthetic grids are still lacking.

## 5. Conclusions

ADM seems to be a material worthy of interest for use in gynaecological surgery. Initial reports on its effectiveness and safety profile are very optimistic. However, more research is needed to fully assess its potential. In particular, the comparison of synthetic meshes and meshes with ADM in laboratory and clinical conditions is a large field for further research. This knowledge would enable the selection of the appropriate material to the individual needs of a particular patient.

## Figures and Tables

**Figure 1 jcm-11-04030-f001:**
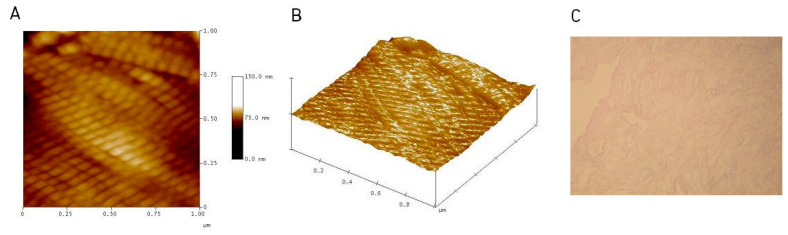
Human acellular dermal matrix (ADM) images: (**A**) atomic force microscopy (AFM), “Height” image; (**B**) atomic force microscopy (AFM), “3D” image; (**C**) H&E image.

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
