# Peer review of "Application of Acellular Dermal Matrix in Gynaecology—A Current Review"

_jcm, 2022, doi:10.3390/jcm11144030_

Round 1
Reviewer 1 Report
Thank you for the opportunity to comment on the manuscript: "Application of acellular dermal matrix in gynaecology – current review". The topic is of potential clinical interest. I support the idea of synthesizing the available data on acellular dermal matrix applications in gynaecology. However, the manuscript as it stands has many significant flaws and needs major improvements before publication. The trouble starts with the Abstract: The first sentence is not informative and should be removed ("Urogynecological disorders are a serious problem for gynaecological care"). The second sentence is not logical and contradictory: “numerous methods” and “this type of operation”. Which type is "this" type? The third sentence is incomplete ("synthetic materials have numerous side effects") and should be supported by 2 or 3 most relevant examples. However, the Abstract is a minor issue. The Methods and Results sections contain basic errors. The Method description is a critical part of any manuscript. Here, it comprises four and half lines. Apparently, the search was limited to two databases. Notably, PubMed is a search engine of MedLine! Most importantly, the reader is not informed about key words, inclusion and exclusion criteria, selection process etc. The authors inform that "The process of cell harvesting from tissues and organs and its importance is a topic that has been described extensively and in detail in the available literature [4-7]". Pointing out to the "available literature" calls into question the purpose of the current review. The basic information should be provided for readers who are less familiar with the subject. What type of review is it? If it is a narrative review, the structure of the manuscript should reflect clearly formulated questions. The information should be provided in a more synthetic, coherent manner. Simply listing selected studies does not make sense. If a systematic approach is intended, the methods and presentation of results are insufficient.Author Response
Thank you for your valuable comments on our article.
Please see the attachment.

Reviewer 2 Report
In this review paper, the use of acellular dermal matrix (ADM) in gynecological treatments has been summarized. This review is interesting ad contributes to the field as there is a need for new materials in gynecological surgeries, especially in pelvic floor reconstruction. However, the research could be more extensive, and more background could be given. The review can be strengthened by addressing the following issues.
Abstract: Abstract is vague, authors should be more specific about the disorders and the treatment methods.
In addition, the aim of the study should be clear and stated early in the abstract.
Line 16-17, please check for the grammar of the sentence.
Line 22-23, the authors should clarify that they can with “ mesh tissue”.
Introduction: The problem and disoırder should be discussed more clearly. What are the disorders and treatment methods? “Urogynecological disorders are a serious problem for gynecological care”, in which terms? Economic, societal, lack of treatment options?
Acellularized Dermal Matrix is indeed an interesting candidate, however, the motive of this study is not clear. Authors should give more background on the disorders, obstacles in treatment, and other materials and tell why ADM instead of many other options. In addition, it would be helpful to explain what is ADM, and what are the sources for ADM.
Materials methods: authors can give examples of the keywords. What type of gynecological disorders have been searched.
Results
Line 61-84 should be reorganized again to better explain what are the important parameters in using acellularized matrixes. I would recommend moving this part to the introduction.
Line 109-113, the details and the conclusion of the study are not clear. “The results confirmed this advantage. Estradiol supplementation increased this effect”, which advantage, which effect?
Line 144 “Very good postoperative results were obtained”, please elaborate on this. What were the results?
Also, in general, there is a lack of information on the follow-up of the studies.
Discussion
Adding limitations of ADM to the search would strengthen the manuscript.
Authors should discuss and compare the properties of the ADM concerning other implants. What are the advantages/disadvantages of the ADM in comparison to other implants/medical devices considering the application in the gynecological field?
It should be stated what should be the future direction and what the authors suggest?
Author Response
Thank you for your valuable comments on our article.
Please see the attachment.

Round 2
Reviewer 1 Report
I'm sorry, but my impression is that a really thorough revision hasn't been done. Resubmitting a manuscript with persistent basic errors raises serious doubts. For example:
- NCBI MedLine, PubMed are still listed separately, while PubMed is a search tool that automatically includes Medline!
- the paragraph related to "this method" and "this type of surgery" has not been improved to make it clear what methods and operations it refers to; - the mansucript still contains remnants of the template: see lines 348-354 "6. Patents";
- the abbreviations are introduced without explanation, e.g. ADM is given without explanation both in the first sentence of the abstract and in the first sentence of the introduction;
- finally, a linguistic revision seems to be mandatory for further processing of the manuscript
Author Response
I'm sorry, but my impression is that a really thorough revision hasn't been done. Resubmitting a manuscript with persistent basic errors raises serious doubts. For example:
- NCBI MedLine, PubMed are still listed separately, while PubMed is a search tool that automatically includes Medline!
The adjustment has been done.
- the paragraph related to "this method" and "this type of surgery" has not been improved to make it clear what methods and operations it refers to;
the terms apply to the entire spectrum of transvaginal surgery, both for improving statics and for reconstructive treatment in the case of congenital abnormalities or complications following oncological treatment. It is impossible to list all types of surgery. The use of ADM in transvaginal surgery is very extensive. Hence the specification of the transvaginal surgeries in the preceding sentence.
- the mansucript still contains remnants of the template: see lines 348-354 "6. Patents";
The redundant paragraph 6 "Patent" has been deleted
- the abbreviations are introduced without explanation, e.g. ADM is given without explanation both in the first sentence of the abstract and in the first sentence of the introduction;
The adjustment has been done.
- finally, a linguistic revision seems to be mandatory for further processing of the manuscript
We have made linguistic and translation improvements.

This manuscript is a resubmission of an earlier submission. The following is a list of the peer review reports and author responses from that submission.